# Catechol-Containing Schiff Bases on Thiacalixarene: Synthesis, Copper (II) Recognition, and Formation of Organic-Inorganic Copper-Based Materials

**DOI:** 10.3390/molecules26082334

**Published:** 2021-04-17

**Authors:** Pavel Padnya, Ksenia Shibaeva, Maxim Arsenyev, Svetlana Baryshnikova, Olga Terenteva, Igor Shiabiev, Artur Khannanov, Artur Boldyrev, Alexander Gerasimov, Denis Grishaev, Yurii Shtyrlin, Ivan Stoikov

**Affiliations:** 1A.M. Butlerov’ Chemistry Institute, Kazan Federal University, 18 Kremlevskaya Street, 420008 Kazan, Russia; alleoks@mail.ru (K.S.); olga-potrekeeva@mail.ru (O.T.); shiabiev.ig@yandex.ru (I.S.); arthann@gmail.com (A.K.); boldyrev25@gmail.com (A.B.); alexander.gerasimov@kpfu.ru (A.G.); 2G.A. Razuvaev Institute of Organometallic Chemistry, Russian Academy of Sciences, 49 Tropinin Street, 603137 Nizhny Novgorod, Russia; mars@iomc.ras.ru (M.A.); baryshnikova@iomc.ras.ru (S.B.); 3Scientific and Educational Center of Pharmaceutics, Kazan Federal University, 420008 Kazan, Russia; dionis.grishaev@yandex.ru (D.G.); yurii.shtyrlin@kpfu.ru (Y.S.)

**Keywords:** thiacalix[4]arenes, Schiff base, synthesis, complexing properties, copper cation, organic-inorganic materials

## Abstract

For the first time, a series of catechol-containing Schiff bases, tetrasubstituted at the lower rim thiacalix[4]arene derivatives in three stereoisomeric forms, *cone*, *partial cone*, and *1,3-alternate*, were synthesized. The structure of the obtained compounds was proved by modern physical methods, such as NMR, IR spectroscopy, and HRMS. Selective recognition (K_b_ difference by three orders of magnitude) of copper (II) cation in the series of d-metal cations (Cu^2+^, Ni^2+^, Co^2+^, Zn^2+^) was shown by UV-vis spectroscopy. Copper (II) ions are coordinated at the nitrogen atom of the imine group and the nearest oxygen atom of the catechol fragment in the thiacalixarene derivatives. High thermal stable organic-inorganic copper-based materials were obtained on the base of *1,3-alternate* + Cu (II) complexes.

## 1. Introduction

The design of new hybrid materials with unusual and unique properties is a promising challenge due to the development of modern industry and the possibilities of their use in advanced technologies [1,2,3,4,5,6,7]. Metals and their ions, such as silver, gold, and copper, are often applied in the composition of organic-inorganic materials, such as an inorganic component [8]. The development of hybrid organic-inorganic materials based on copper ions is promising due to the combination of their unique properties and low cost [9,10,11]. Such materials have high thermal and electrical conductivity, biological activity (for example, antifungal, antibacterial), and are also used as catalysts and sensors [12,13,14,15]. 

Schiff bases are one of the most convenient and effective complexing compounds [16,17,18]. They are interesting due to a number of reasons, e.g., (1) easy synthesis by condensation of aldehydes or ketones to primary amines under mild conditions; (2) the possibility of designing and changing the properties of target structures by varying the substituents in the starting amine and aldehyde (ketone). There are a large number of examples of the creation of promising materials based on the Schiff bases [19,20,21,22,23]. However, their low selectivity is a main problem of such complexing and chelating agents.

Synthetic macrocyclic compounds, such as calixarenes, thiacalixarenes, resorcinarenes, pillararenes, and others, have excellent complexing properties and are often selective for a certain substrate [24,25,26,27,28,29,30,31,32]. The introduction of fragments of Schiff bases into the structure of (thia)calixarenes makes it possible to increase both the efficiency and selectivity with respect to metal cations.

At the moment, there are many examples of the synthesis of Schiff bases based on (thia)calixarene derivatives, as well as the study of complexing properties, with respect to a series of metal cations, including copper (II) cations [33,34,35,36,37]. However, most of the obtained macrocyclic Schiff bases do not possess selectivity toward *d*-element cations, such as Cu^2+^, Ni^2+^, Co^2+^, and Zn^2+^. The introduction of amide groups capable of spatial pre-organization of the receptor fragments via additional hydrogen bonds into the macrocycle structure can lead to a change in the complexing properties and improvement in the selectivity of the resulting molecules.

In this work, Schiff bases with catechol fragments on thiacalix[4]arenes substituted at the lower rim in three conformations (*cone*, *partial cone*, and *1,3-alternate*) were synthesized for the first time. Selective recognition of copper (II) cations in the series of *d*-element cations (copper (II), nickel (II), cobalt (II), and zinc (II)) was shown by UV-vis spectroscopy. Organic-inorganic copper-based materials were obtained on the base of thiacalixarene-copper (II) complexes, and their properties were studied.

## 2. Results and Discussion

### 2.1. Synthesis of thiacalix[4]arene-Based Schiff Bases

Previously, in the group of Professor Hosseini, catechol-containing Schiff bases, tetrasubstituted at the lower rim calix[4]arene and thiacalix[4]arene (L) in *1,3-alternate* conformation, were synthesized [37]. These compounds were capable of complexation of metal (II) ions (Cu (II), Co (II), Ni (II), and Zn (II)) with low selectivity. The authors have shown the formation of complexes with the composition M/L = 2/1 in the CH_3_OH/CHCl_3_ (1:1) solution and in the crystal in case of the macrocycle-Cu (II) complexes.

It is known that the arrangement of substituents of (thia)calixarenes strongly affects complexing properties of the resulting receptors in relation to various substrates (metal cations, anions, small organic molecules, proteins, nucleic acids) [38,39,40,41,42,43]. The macrocycle acts as a platform capable of rigid spatial arrangement of receptor fragments. Therefore, we proposed to synthesize new Schiff bases on thiacalix[4]arene derivatives in three different stereoisomeric forms, *cone*, *partial cone*, and *1,3-alternate*, and also to study their complexing properties with metal cations to obtain new organic-inorganic nanomaterials. Easily obtainable tetraester derivatives **2**–**4** were proposed as precursors of desired stereoisomers. Spatial fixation of ester groups in space, without the possibility of conversion of stereoisomeric forms into each other without breaking chemical bonds, was the advantage of these compounds. Further, tetrasubstituted thiacalix[4]arenes **5**–**7**, containing both amide and primary amine groups separated by a hexamethylene spacer, were synthesized in three conformations, *cone*, *partial cone*, and *1,3-alternate*, according to the method developed in our research group [44] (Scheme 1).

Further reaction of the compounds **5**–**7** with 4,6-di-*tert*-butyl-1,2-dihydroxybenzaldehyde in methanol at room temperature gave Schiff bases **8**–**10** (Figure 1) in high yields (91–96%). 4,6-Di-*tert*-butyl-1,2-dihydroxybenzaldehyde is a convenient synthetic block for the synthesis of oligomeric Schiff bases with catechol or *o*-quinone fragments, redox-active ligands for coordination chemistry. Due to the presence of *tert*-butyl groups, it is possible to isolate corresponding *o*-benzoquinones and register the oxidized forms of catecholate complexes. This aldehyde also has the potential to act as both an *O*,*O*-chelating moiety and an *O*,*N*-chelating moiety [45,46,47]. Unlike vanillin derivatives, it can participate in the formation of dimeric structures via intermolecular hydrogen bonds. Monomeric compound **11** (Figure 1) was synthesized with a yield of 90% in a similar synthetic way to compare complexing properties of the obtained macrocyclic compounds. The structure and composition of the obtained compounds **8**–**11** were confirmed by modern physical methods, e.g., ^1^H, ^13^C NMR, IR spectroscopy, high-resolution mass spectrometry (HRMS), and elemental analysis (Appendix A).

Two-dimensional NMR spectroscopy is a common technique for establishing the structure and conformation of macromolecules. However, it is known that different conformations of thiacalixarene derivatives can be unequivocally confirmed by ^1^H NMR spectroscopy data of specific proton signals. The conformational differentiation of *cone* and *1,3-alternate* stereoisomers of amide derivatives of *p-tert*-butylthiacalix[4]arenes tetrasubstituted at the lower rim can be performed by chemical shifts of the *tert*-butyl group, aromatic ring, oxymethylene, amide, and amidomethylene protons in the ^1^H NMR spectra (Appendix A). In the case of compound **10** (*1,3-alternate*), the protons of the OCH_2_, NHCH_2_, and amide groups were located in the zone shielded by neighboring aromatic rings of the macrocycle. Their signals were recorded at stronger fields in the ^1^H NMR spectrum (4.06, 3.24, and 7.75 ppm, respectively) than those of the protons of the macrocycle **8** in *cone* conformation (4.80, 3.33, and 7.93 ppm, respectively). Chemical shifts of the aromatic protons depended less on the conformation of the macrocyclic cavity. They drifted upfield by only 0.19 ppm from *cone*
**8** (7.33 ppm) to *1,3-alternate*
**10** (7.52 ppm) stereoisomers. This was evidence of the shielding effect of neighboring aryl fragments in *cone* stereoisomer on the aryl protons of macrocycle ring. The signals of the *tert*-butyl group protons of *cone*
**8** were found at a stronger field (1.09 ppm) against corresponding proton signals of the *1,3-alternate* stereoisomer **10** (1.18 ppm). This was probably due to the spatial location of the *tert*-butyl groups of the *1,3-alternate* stereoisomer shielded by neighboring fragments of the macrocycle.

However, the spectrum of compound **9** in *partial cone* conformation was complicated by the decrease of the structure symmetry. The signals of the *tert*-butyl protons appeared as three singlets (1.02, 1.26, 1.29 ppm), with an integral intensity ratio of 2:1:1. Signals of the amide protons appeared as three broad triplets (7.53, 7.88, and 8.64 ppm), with an integral intensity ratio of 1:2:1. Meanwhile, the signals of the oxymethylene and aromatic protons formed two singlets and an AB-system, respectively (Appendix A).

It is interesting to note that chemical shifts of the signals of the remaining protons were almost independent of the macrocycle conformation. They were close to the chemical shifts of the signals of the same protons in the monomer **11** (Appendix A). It is likely that their effect did not play a significant role at a certain distance from the thiacalixarene platform.

The structure of the obtained Schiff bases **8**–**11** was also studied by FT-IR spectroscopy (Appendix A). One of the main significant absorption bands of the C=N bond was in the range of 1616–1618 cm^−1^ in the case of the macrocyclic compounds **8**–**10** and 1623 cm^−1^ in the case of the monomeric compound **11**.

The obtained compounds **8**–**11** were also characterized by HRMS (Appendix A). Mass spectra of the macrocyclic compounds **8**–**10** showed peaks corresponding to molecular ions with two, three, and four protons ([M + 2H]^2+^, [M + 3H]^3+^, and [M + 4H]^4+^). In the case of the monomeric compound **11**, a peak (*m*/*z* = 539.3849) corresponding to the protonated molecular ion [M + H]^+^ was recorded.

Thus, new Schiff bases, tetrasubstituted at the lower rim thiacalixarenes containing catechol fragments in the structure, were obtained in high yields. The structure of the complexes obtained was proven by modern physical methods.

### 2.2. The Study of Complexing Properties of the Obtained thiacalix[4]arene-Based Schiff Bases

Complexing ability of the obtained Schiff bases **8**–**11** with divalent *d*-metal cations (copper (II), zinc (II), cobalt (II), and nickel (II)) was studied by UV-vis spectroscopy in the CH_3_OH-CHCl_3_ mixture (1:1) at room temperature. Figure 2 and Appendix A show the UV-vis spectra of the mixtures of thiacalixarenes **8**–**10** and the monomer **11** with a 10-fold excess of all the studied metal cations in the mixture CH_3_OH-CHCl_3_ (1:1). UV-vis spectra of the macrocyclic compounds **8**–**10** are typical for catecholaldimines in polar solvents [48]. There are four main bands in the UV spectra of pure macrocyclic compounds **8**–**10**. Two bands at 250 and 285 nm are assigned to π → π *– transitions of the macrocyclic main chains, a band at 320 nm is assigned to n → π *– transitions of the C=N imine groups, while a broad band at 430 nm corresponds to intramolecular charge transfer.

Changes in the electronic spectra of the thiacalix[4]arenes **8**–**10** were found only in the case of the addition of the copper (II) solution. In this case, the absorption band in the region of 320 nm disappeared and a hypsochromic (blue) shift of the absorption band at 430 nm to the region of 400 nm was observed. The appearance of new bands corresponded to the formation of new coordination bonds between the copper (II) cations and the ligands. An increase in absorption in the region of 520 nm was also observed, so that color of the solution changed from yellow-orange to orange-red when adding the Cu^2+^ ions (Figure 2).

Complexing properties of the macrocyclic compounds **8**–**10** were compared with those of the monomeric compound **11**. Appendix A shows UV-vis spectra of the compound **11** without/with a 10-fold excess of all the studied metal cations in the CH_3_OH-CHCl_3_ mixture (1:1). One can see that compound **11** interacted with all the cations (copper (II), zinc (II), cobalt (II), and nickel (II)) and did not exhibit any selectivity.

Further, binding constants and stoichiometry of the complexes of the copper (II) cation with the compounds **8**–**11** were determined. Stoichiometry was determined by Job’s plot. Figure 3B and Appendix A show the results of complexation of the compounds **8**–**11**. In the case of the macrocyclic compounds **8**–**10**, maximum value of the molar fraction was 0.5, corresponding to the 1:1 stoichiometry of the macrocycle-Cu (II). In the case of the monomeric compound **11**, maximum molar fraction corresponded to the value close to 0.33, which corresponded to the 2:1 stoichiometry of compound **11**/Cu (II).

Values of the binding constants of the compounds **8**–**11** with the copper (II) ions were measured by the UV-vis titration (Figure 3A, Appendix A, Table 1). Using spectroscopic data obtained for the complexes of the compounds **8**–**11** with the copper (II) cations, binding constants (K_b_) were calculated for all four complexes using the Bindfit software [49]. The stoichiometry of the complexes of the macrocycles **8**–**10** (1:1) and the monomer compound **11** (2:1) was additionally confirmed by Bindfit (Appendix A.). The calculated values of the logK_b_ of the macrocyclic compounds **8**–**10** were close to each other in the range of 5.04–5.46. Such low binding constants of the macrocyclic compounds **8**–**10** in comparison with the monomeric compound **11** (logK’_b_ = 8.52) can be explained by steric preorganization of the complexing fragments relative to the thiacalixarene platform. This lead to less flexibility in the macrocyclic structures. However, this preorganization lead to the selectivity of recognition of the copper (II) cation by thiacalixarenes **8**–**10** in the series of *d*-metal cations in comparison with the monomeric compound **11**, which bound all the studied cations.

It is known that changes in the UV-vis spectra may not be observed at low binding constants (K_b_ ≤ 100) [50]. Based on the data, we asserted that the selectivity of the recognition of the copper (II) ions by the obtained macrocyclic compounds **8**–**10** was at least 1000 (K_b_ difference by three orders of magnitude).

It was shown earlier by Hosseini [37] that the coordination of the copper (II) ions occurs with two fragments of the (thia)calix[4]arene substituents. In this case, coordination occurred at the nitrogen atom of the C=N bond and nearest the oxygen atom of the catechol fragment. In our case, binding constants for the complexes of the macrocycles **8**–**10** with the copper (II) cations (and the stoichiometry is 1:1) were close (log K_b_ = 5.04–5.46), while the stoichiometry of the complex of the monomeric compound with Cu (II) was 2:1. Such results can be explained by the fact that two substituents located oppositely each other were coordinated by the copper (II) ions, while the other two substituents did not participate in the complexation. Inability of free substituents to complexation can be explained by the allosteric effect. Change of the conformation of the macrocyclic platform occurred upon the complexation of one copper ion, and the coordination of the second copper (II) cation became impossible. Figure 4 schematically shows the complexation of the macrocycles **8**–**10** and the monomer **11** with the copper (II) cation.

However, our results are fundamentally different from the previous work of professor Hosseini [37]. Here, introduction of the imine groups with the catechol fragments into the macrocyclic platform of thiacalix[4]arene led to the creation of selective receptors (K_b_ difference by three orders of magnitude) for the copper (II) cations in a series of *d*-metal cations (Cu^2+^, Co^2+^, Ni^2+^, and Zn^2+^). Complexes with the stoichiometry of 1:1 were formed in our case.

It is known that the copper (II) cation is paramagnetic [51]. Therefore, the study of complexation by the NMR spectroscopy can often be difficult. We decided to study the interaction of compounds **8**–**11** with the deficiency of the copper ions to confirm coordination of the cations. Figure 5 and Appendix A show ^1^H NMR spectra of compounds **8**–**11** (10 mM, CD_3_OD:CDCl_3_ = 1:1), with the addition of different amounts of the copper (II) cations. Chemical shifts of the proton signals at 8.70 ppm (N=CH), 6.44 ppm (ArH_catechol_), and 3.51 ppm (CH_2_N=C) shifted downfield by 0.15–0.20 ppm (Figure 5). Such changes unambiguously confirmed the coordination of the copper (II) ions at the nitrogen atom of the C=N group and the nearest oxygen atom of the catechol fragment. Small changes in the chemical shifts of the protons of the aromatic fragments and the *tert*-butyl groups of the macrocyclic platform (0.02–0.04 ppm) confirmed hypothesis of the changes in the conformation of the macrocycles upon complexation.

Thus, the studied macrocyclic compounds **8**–**10** were selective receptors for the copper (II) cation among the cations of other divalent *d*-metals. The main reason for this selectivity is associated with steric preorganization of coordination groups (C=N groups and catechol fragments) by the macrocyclic thiacalixarene platform. In this case, the logarithms of the binding constants turned out to be close.

### 2.3. Synthesis and Characterization of Thiacalixarene-Copper (II) Materials

Design of thermally stable materials is a promising and popular direction for the creation of new optical, photoelectric, and electrochemically active devices. The use of copper-based organic-inorganic hybrid materials is convenient due to good target properties and low cost of copper against commonly used metals, such as silver and gold [9,10,11,12]. Due to high selectivity of the studied compounds **8**–**10** to the copper (II) cation in the series of *d*-metal cations, we proposed to obtain organic-inorganic materials based on them and to study their properties.

Thiacalixarene **10** in *1,3-alternate* conformation with the highest binding constant was used as an organic component to create organic-inorganic materials. Solution of the copper (II) chloride (1 mM, CH_3_OH-CHCl_3_ (1:1)) was added to the yellow solution (1 mM) of the thiacalixarene **10** in the mixture CH_3_OH:CHCl_3_ (1:1) at room temperature. Concentration and evaporation of the solution gave orange-red amorphous powder (**10** + Cu complex), which was dried in vacuum for 48 h. Unfortunately, we failed to obtain the crystalline sample. Starting thiacalixarene **10** and copper (II) chloride were isolated under the same conditions. Mechanical mixture of the thiacalixarene-CuCl_2_ (**10** + Cu mixture) was also prepared to compare the properties of the obtained organic-inorganic material.

Initially, FT-IR spectroscopy was used to determine the complexation behavior of compound **10** with the copper (II) cation (Figure 6 and Appendix A). In the FT-IR spectrum of compound **10**, the C=N bond band with the wavenumber of 1618 cm^−1^ and the amide group band at 1668 cm^−1^ were characteristic. In the obtained material **10** + Cu complex, bifurcation of the ν(C=N) stretching band (1627 and 1595 cm^−1^) was observed, as well as the shift of the absorption band of the amide group (up to 1655 cm^−1^). The latter can be explained by the allosteric effect of the interaction of compound **10** with the copper (II) ions described in Section 2.2. There was convergence of the catechol fragments of the Schiff bases on one side of the macrocyclic platform as a result of complexation with the copper ion. Meanwhile, the distance between the substituents increased on the other side. Therefore, two bands of the C=N bond appeared to correspond to the C=N fragment (1595 cm^−1^) bound to the copper (II) ion, as well as free substituents with Schiff base fragments (1627 cm^−1^). Schematic representation of the coordination is shown in Figure 4. It should be noted that the spectrum of the mechanical mixture thiacalixarene-CuCl_2_ (**10** + Cu mixture) corresponded to the sum of the spectra of initial thiacalixarene **10** and CuCl_2_. Shift of the absorption bands and the appearance of new bands were also not observed in this case.

Further, the obtained samples were studied by the PXRD method. Appendix A shows PXRD spectra of compound **10**, CuCl_2_, their mechanical mixture (**10** + Cu mixture), and our material (**10** + Cu complex). Since the studied samples were isolated from the CH_3_OH:CHCl_3_ mixture, we observed peaks corresponding to a mixture of CuCl_2_ hydrate and methanolate structures in the case of a copper salt. It is worth noting the lack of crystallinity of the macrocycle **10**, as well as the resulting organic-inorganic material (**10** + Cu complex). One can see that the PXRD spectrum of the obtained material differed from that of the mechanical mixture (**10** + Cu mixture). This confirmed the complexation of the copper (II) cations with the organic ligand.

The obtained samples were also studied using the TG method (Figure 7 and Appendix A). It turned out that compound **10** in its pure form was thermally stable (decomposition began at a temperature of 314 °C). The resulting organic-inorganic material **10** + Cu complex was more thermally stable (up to 326 °C) against compound **10**.

Thus, as a result of this work, we synthesized Schiff bases with catechol fragments based on thiacalix[4]arenes, substituted at the lower rim in three conformations, *cone*, *partial cone*, and *1,3-alternate*. Selective recognition (K_b_ difference by three orders of magnitude) of the copper (II) cations in the series of *d*-element cations (copper (II), nickel (II), cobalt (II), and zinc (II)) was shown by UV-vis spectroscopy. It was demonstrated by a number of physical methods that the copper (II) ions coordinated at the nitrogen atom of the imine group and the nearest oxygen atom of the catechol fragment in the thiacalixarene derivatives. Thermally highly stable organic-inorganic copper-based materials were obtained on the base of thiacalixarene-copper (II) complexes.

## 3. Materials and Methods

### 3.1. General

All chemicals were purchased from Acros (Fair Lawn, NJ, USA), and most of them were used as received without additional purification. Organic solvents were purified by standard procedures. ^1^H NMR and ^13^C NMR spectra were obtained on a Bruker Avance-400 spectrometer (Bruker Corp., Billerica, MA, USA) (^13^C{^1^H} 100 MHz and ^1^H 400 MHz). Chemical shifts were determined against the signals of residual protons of deuterated solvent (CDCl_3_). The concentrations of the compounds were equal to 3–5% in all the records. The FTIR ATR spectra were recorded on the Spectrum 400 FT-IR spectrometer (Perkin Elmer, Seer Green, Lantrisant, UK) with a Diamond KRS-5 attenuated total internal reflectance attachment (resolution 0.5 cm^−1^, accumulation of 64 scans, recording time 16 s in the wavelength range 400–4000 cm^−1^). HRMS mass spectra were obtained on a quadrupole time-of-flight (t, qTOF) AB Sciex Triple TOF 5600 mass spectrometer (AB SCIEX PTE. Ltd., Singapore) using a turbo-ion spray source (nebulizer gas nitrogen, a positive ionization polarity, needle voltage 5500 V). Recording of the spectra was performed in “TOF MS” mode with collision energy 10 eV, declustering potentially 100 eV and with a resolution of more than 30,000 full-width half-maximum. Samples with the analyte concentration of 5 μmol/L were prepared by dissolving the test compounds in the mixture of methanol (HPLC-UV Grade, LabScan). Elemental analysis was performed on Perkin–Elmer 2400 Series II instruments (Perkin Elmer, Waltham, MA, USA). Melting points were determined using Boetius Block apparatus (VEB Kombinat *Nagema*, Radebeul, Germany). Thermogravimetry (TG) analysis of solid samples were performed using the thermoanalyzer STA 449F1 Jupiter (Netzsch, Germany) at the temperature range of 40–550 °C. In each experiment, the temperature scanning rate was 10 °C/min, and an argon with a total flow rate of 75 mL/min was used. Powder X-ray diffraction (PXRD) studies were performed using a MiniFlex 600 diffractometer (Rigaku, Japan) equipped with a D/teX Ultra detector. Cu Kα radiation (40 kV, 15 mA) was used for all the measurements, which were collected at room temperature in the range of 2θ from 3 to 100° with 0.02° steps and 0.24 s exposure time at each point without sample rotation. UV-vis spectra were recorded on a Shimadzu UV-3600 spectrometer (Kyoto, Japan). Quartz cuvettes with an optical path length of 10 mm were used. A mixture of CH_3_OH:CHCl_3_ (1:1) was used for preparation of the solutions. Absorption spectra of mixtures were recorded after 1 hour of incubation at 20 °C.

Thiacalix[4]arenes **5**–**7** were synthesized according to the literature procedure [44] and 4,6-di-*tert*-butyl-1,2-dihydroxybenzaldehyde according to [52].

### 3.2. General Procedure for the Synthesis of the Compounds ***8***–***10***

A mixture of 4,6-di-*tert*-butyl-1,2-dihydroxybenzaldehyde (148.8 mg, 0.1486 mmol) and thiacalix[4]arene-based amines **5**–**7** (200 mg, 0.1486 mmol) in methanol (5 mL) was stirred at room temperature. The precipitate formed was then filtrated. The residue was washed with methanol (3 × 15 mL), isolated as orange crystalline powders, and dried under reduced pressure over phosphorus pentoxide.

#### 3.2.1. 5,11,17,23-Tetra-*tert*-butyl-25,26,27,28-tetrakis[*N*-(6-(4,6-di-*tert*-butyl-2,3-dihydroxybenzylideneamino)hexyl)carbamoylmethoxy]-2,8,14,20-tetrathiacalix[4]arene in *cone* Conformation (**8**)

Yellow-orange powder, yield: 314 mg (92%). M.p. 110 °C. ^1^H NMR (CDCl_3_, *δ*, ppm, *J*/Hz): 1.09 (s, 36H, (CH_3_)_3_C), 1.38 (s, 36H, (CH_3_)_3_C), 1.39 (s, 36H, (CH_3_)_3_C), 1.35–1.42 (m, 16H, NH(CH_2_)_2_(CH_2_)_2_(CH_2_)_2_N=CH), 1.59 (m, 8H, NHCH_2_
CH_2_(CH_2_)_4_N=CH), 1.70 (m, 8H, NH(CH_2_)_4_CH_2_CH_2_N=CH), 3.33 (m, 8H, NHCH_2_(CH_2_)_5_N=CH), 3.56 (m, 8H, NH(CH_2_)_5_CH_2_N=CH), 4.80 (s, 8H, OCH_2_CO), 6.52 (s, 4H, ArH(C_6_H_1_)), 7.33 (s, 8H, ArH), 7.92 (br.t, 4H, NH), 8.80 (s, 4H, CH=N). ^13^C NMR (CDCl_3_, *δ*, ppm): 168.4, 164.1, 157.8, 147.6, 143.9, 135.0, 128.1, 115.3, 113.3, 110.0, 74.6, 54.0, 39.2, 35.6, 34.4, 33.8, 33.4, 32.5, 31.2, 30.3, 29.4, 29.2, 29.0, 26.4, 26.3. FTIR ATR (*ν*, cm^−1^): 3326 (NH), 3061 (NH), 1671 (C(O)NH, amide I), 1618 (C=N), 1532 (C(O)NH, amide II), 1092 (C_Ar_OCH_2_). Elemental analysis. Calculated for C_132_H_192_N_8_O_16_S_4_ C, 66.74; H, 8.44; N, 6.77; S, 7.75; Found: C, 66.68; H, 8.16; N, 6.71; S, 7.39. HRMS: calculated [M + 2H]^2+^
*m*/*z* = 1137.6743, [M + 3H]^3+^
*m*/*z* = 758.7852, [M + 4H]^4+^
*m*/*z* = 569.3408, found [M + 2H]^2+^
*m*/*z* = 1137.6723, [M + 3H]^3+^
*m*/*z* = 758.7846, [M + 4H]^4+^
*m*/*z* = 569.3409.

#### 3.2.2. 5,11,17,23-Tetra-*tert*-butyl-25,26,27,28-tetrakis[*N*-(6-(4,6-di-*tert*-butyl-2,3-dihydroxybenzylideneamino)hexyl)carbamoylmethoxy]-2,8,14,20-tetrathiacalix[4]arene in *partial cone* Conformation (**9**)

Yellow-orange powder, yield: 308 mg (91%). M.p. 117 °C. ^1^H NMR (CDCl_3_, *δ*, ppm, *J*/Hz): 1.02 (s, 18H, (CH_3_)_3_C), 1.26 (s, 9H, (CH_3_)_3_C), 1.29 (s, 9H, (CH_3_)_3_C), 1.37 (s, 36H, (CH_3_)_3_C), 1.41 (s, 36H, (CH_3_)_3_C), 1.42–1.48 (m, 16H, NH(CH_2_)_2_(CH_2_)_2_(CH_2_)_2_N=CH), 1.59–1.67 (m, 8H, NHCH_2_
CH_2_(CH_2_)_4_N=CH), 1.72–1.76 (m, 8H, NH(CH_2_)_4_CH_2_CH_2_N=CH), 3.27–3.42 (m, 8H, NHCH_2_(CH_2_)_5_N=CH), 3.61 (m, 8H, NH(CH_2_)_5_CH_2_N=CH), 4.26 (d of AB system, 2H, OCH_2_CO, *^2^**J_HH_* = 14.4), 4.86 (d of AB system, 2H, OCH_2_CO, *^2^**J_HH_* = 14.4), 4.44 (s, 2H, OCH_2_CO), 4.96 (s, 2H, OCH_2_CO), 6.53 (s, 4H, ArH(C_6_H_1_)), 7.05 (d of AB system, 2H, ArH, *^4^**J_HH_* = 2.4), 7.45 (d of AB system, 2H, ArH, *^4^**J_HH_* = 2.4), 7.53 (br.t, 1H, NH), 7.61 (C, 2H, ArH), 7.76 (C, 2H, ArH), 7.88 (br.t, 2H, NH), 8.64 (br.t, 1H, NH), 8.83 (s, 4H, CH=N). ^13^C NMR (CDCl_3_, *δ*, ppm): 168.8, 168.5, 168.1, 163.9, 159.4, 158.0, 155.9, 147.6, 146.6, 143.8, 139.9, 139.8, 139.8, 136.4, 135.2, 135.1, 133.4, 127.9, 127.4, 125.9, 125.8, 112.5, 109.7, 74.3, 73.8, 70.2, 54.5, 54.3, 54.3, 39.3, 39.2, 38.9, 35.5, 35.3, 33.3, 31.4, 31.2, 31.1, 30.6, 30.5, 29.7, 29.6, 29.2, 26.9, 26.6, 26.5, 26.5, 26.4. FTIR ATR (*ν*, cm^−1^): 3312 (NH), 3062 (NH), 1673 (C(O)NH, amide I), 1616 (C=N), 1532 (C(O)NH, amide II), 1088 (C_Ar_OCH_2_). Elemental analysis. Calculated for C_132_H_192_N_8_O_16_S_4_ C, 66.74; H, 8.44; N, 6.77; S, 7.75; Found: C, 66.72; H, 8.54; N, 6.43; S, 7.79. HRMS: calculated [M + 2H]^2+^
*m*/*z* = 1137.6743, [M + 3H]^3+^
*m*/*z* = 758.7852, [M + 4H]^4+^
*m*/*z* = 569.3408, found [M + 2H]^2+^
*m*/*z* = 1137.6745, [M + 3H]^3+^
*m*/*z* = 758.7857, [M + 4H]^4+^
*m*/*z* = 569.3418.

#### 3.2.3. 5,11,17,23-Tetra-*tert*-butyl-25,26,27,28-tetrakis[*N*-(6-(4,6-di-*tert*-butyl-2,3-dihydroxybenzylideneamino)hexyl)carbamoylmethoxy]-2,8,14,20-tetrathiacalix[4]arene in *1,3-alterate* Conformation (**10**)

Yellow-orange powder, yield: 324 mg (96%). M.p. 115 °C. ^1^H NMR (CDCl_3_, *δ*, ppm, *J*/Hz): 1.18 (s, 36H, (CH_3_)_3_C), 1.38 (s, 36H, (CH_3_)_3_C), 1.41 (s, 36H, (CH_3_)_3_C), 1.35–1.43 (m, 16H, NH(CH_2_)_2_(CH_2_)_2_(CH_2_)_2_N=CH), 1.60 (m, 8H, NHCH_2_
CH_2_(CH_2_)_4_N=CH), 1.74 (m, 8H, NH(CH_2_)_4_CH_2_CH_2_N=CH), 3.24 (m, 8H, NHCH_2_(CH_2_)_5_N=CH), 3.61 (m, 8H, NH(CH_2_)_5_CH_2_N=CH), 4.06 (s, 8H, OCH_2_CO), 6.60 (s, 4H, ArH(C_6_H_1_)), 7.52 (s, 8H, ArH), 7.75 (br.t, 4H, NH), 8.84 (s, 4H, CH=N). ^13^C NMR (CDCl_3_, *δ*, ppm): 168.2, 164.0, 156.9, 147.5, 143.8, 133.3, 127.2, 115.3, 112.8, 110.1, 71.2, 54.6, 39.4, 35.6, 35.3, 34.4, 33.8, 33.4, 31.2, 30.6, 29.6, 29.2, 29.0, 26.8, 26.5. FTIR ATR (*ν*, cm^−1^): 3319 (NH), 3065 (NH), 1668 (C(O)NH, amide I), 1618 (C=N), 1531 (C(O)NH, amide II), 1085 (C_Ar_OCH_2_). Elemental analysis. Calculated for C_132_H_192_N_8_O_16_S_4_ C, 66.74; H, 8.44; N, 6.77; S, 7.75; found: C, 66.52; H, 8.64; N, 6.63; S, 7.74. HRMS: calculated [M + 2H]^2+^
*m*/*z* = 1137.6743, [M + 3H]^3+^
*m*/*z* = 758.7852, [M + 4H]^4+^
*m*/*z* = 569.3408, found [M + 2H]^2+^
*m*/*z* = 1138.1733, [M + 3H]^3+^
*m*/*z* = 758.7826, [M + 4H]^4+^
*m*/*z* = 569.3399.

### 3.3. Procedure for the Synthesis of Compound ***11*** (Monomer)

A mixture of 4,6-di-*tert*-butyl-1,2-dihydroxybenzaldehyde (163.4 mg, 0.6527 mmol) and *N*-(6-aminohexyl)-2-(4-*tert*-butylphenoxy)acetamide [44] (200 mg, 0.6527 mmol) in methanol (5 mL) was stirred at room temperature. The precipitate formed was then filtrated. The residue was washed with methanol (3 × 15 mL), isolated as orange crystalline powders, and dried under reduced pressure over phosphorus pentoxide.

#### 2-(4-*tert*-Butylphenoxy)-*N*-(6-(4,6-di-*tert*-butyl-2,3-dihydroxybenzylideneamino)hexyl)acetamide (**11**)

Yellow-orange powder, yield: 316 mg (90%). M.p. 55 °C. ^1^H NMR (CDCl_3_, *δ*, ppm, *J*/Hz): 1.29 (s, 9H, (CH_3_)_3_C), 1.40 (s, 9H, (CH_3_)_3_C), 1.42 (s, 9H, (CH_3_)_3_C), 1.33–1.48 (m, 4H, NHCH_2_CH_2_(CH_2_)_2_(CH_2_)_2_N=CH), 1.56 (m, 2H, NHCH_2_CH_2_(CH_2_)_4_N=CH), 1.72 (m, 2H, NH(CH_2_)_4_CH_2_CH_2_N=CH), 3.34 (m, 2H, NHCH_2_(CH_2_)_5_N=CH), 3.58 (m, 2H, NH(CH_2_)_5_CH_2_N=CH), 4.46 (s, 2H, OCH_2_CO), 6.56 (s, 1H, ArH(C_6_H_1_)), 6.63 (br.t, 1H, NH), 6.85(m, 2H, ArH), 7.33 (m, 2H, ArH), 8.82 (s, 1H, CH=N). ^13^C NMR (CDCl_3_, *δ*, ppm): 168.5, 163.8, 157.8, 155.0, 148.1, 143.7, 136.2, 126.4, 114.2, 112.3, 109.8, 67.5, 54.2, 38.8, 35.5, 35.2, 34.2, 33.8, 33.3, 31.5, 30.5, 29.5, 29.2, 29.1, 26.4, 26.3. FTIR ATR (*ν*, cm^−1^): 3298 (NH), 3068 (NH), 1666 (C(O)NH, amide I), 1623 (C=N), 1511 (C(O)NH, amide II), 1058 (C_Ar_OCH_2_). Elemental analysis. Calculated for C_3__3_H_50_N_2_O_4_ C, 73.57; H, 9.35; N, 5.20; Found: C, 73.42; H, 9.25; N, 5.43. HRMS: calculated [M + H]^+^
*m*/*z* = 539.3843, found [M + H]^+^
*m*/*z* = 539.3849.

### 3.4. Job’s Plots

A series of the solutions of compounds **8**–**11** and CuCl_2_ × 2H_2_O were prepared in CH_3_OH:CHCl_3_ (1:1). The volume ratio of the host and guest solutions varied from 0.3:2.7 to 2.7:0.3, respectively, with the total concentration of the host (H) and guest (G) being constant and equal to 3 × 10^−5^ M. The solutions were used without further stirring. The absorbance A_i_ of the complexation systems was measured at the maximum absorbance wavelength of the complex. The absorbance values were used to plot a diagram from which maximum the structures of the complexes were deduced. Three independent experiments were carried out for each system.

### 3.5. Determination of the Stability Constant and Stoichiometry of the Complex by Spectrophotometric Titration

The 1 × 10^−5^ M (3 × 10^−5^ M in the case of **11**) solution of the Cu^2+^ (300, 600, 900, 1200, 1500, 1800, 2100, 2400, and 2700 μL) in CHCl_3_:MeOH (1:1) was added to 0.3 mL of the solution of **8**–**10** (1 × 10^−4^ M) or **11** (3 × 10^−4^ M) in CHCl_3_:MeOH (1:1) and diluted to the final volume of 3 mL with CHCl_3_-MeOH (1:1). The UV-vis spectra of the solutions were then recorded. The stability constants of complexes were calculated by Bindfit. Three independent experiments were carried out for each series.

### 3.6. Procedure for the Synthesis of Thiacalixarene ***10***—Copper (II) Materials

A solution of copper (II) chloride (1 mM, CH_3_OH:CHCl_3_ (1:1)) was added to a yellow solution (1 mM) of thiacalixarene **10** in a mixture CH_3_OH:CHCl_3_ (1:1) at room temperature. Concentration and evaporation of the solution gave amorphous orange-red powder (**10** + Cu complex) with yield 95%. This powder was dried in vacuum for 48 h.

## 4. Conclusions

For the first time, a series of catechol-containing Schiff bases, tetrasubstituted on the lower rim thiacalix[4]arene derivatives in three stereoisomeric forms, *cone*, *partial cone* and *1,3-alternate*, were synthesized with high yields. The structure of the obtained compounds was proved using a number of modern physical methods, such as NMR, IR, UV-vis spectroscopy, and high-resolution mass spectrometry. Selective recognition (K_b_ difference by three orders of magnitude) of the copper (II) cations in the series of *d*-metal cations (copper (II), nickel (II), cobalt (II), and zinc (II)) was shown by UV-vis spectroscopy. The logarithms of binding constants were found to be similar for all three stereoisomers (*cone* + Cu (II): 5.04; *partial cone* + Cu (II): 5.36; *1,3-alternate* + Cu (II): 5.46). It was shown by a series of physical methods, that the copper (II) ions were coordinated at the nitrogen atom of the imine group and the nearest oxygen atom of the catechol fragment in thiacalixarene derivatives. Thermally stable organic-inorganic copper-based materials were obtained on the base of *1,3-alternate* + Cu (II) complexes. The obtained materials can find application as antifungal and antibacterial coatings and can also be used as catalysts or in the assembly of chemical and electrochemical sensors.

## Data Availability

The data presented in this study are available in Appendix A.

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
