# Peer review of "Catechol-Containing Schiff Bases on Thiacalixarene: Synthesis, Copper (II) Recognition, and Formation of Organic-Inorganic Copper-Based Materials"

_molecules, 2021, doi:10.3390/molecules26082334_

Round 1

Reviewer 1 Report

The paper deals with synthesis of new calixarene derivatives as Schiff bases and their complexation properties towards Cu(II). Compounds were analyzed by usual analytical methods (NMR, IR, UV-Vis, CHN); stoichiometry and stability constants were determined by UV-Vis spectroscopy, selectivity was achieved regarding Ni(II), Co(II) and Zn(II) metal ions; results are well discussed and adequate conclusions are given.

I recommend the publication after minor revision.

Some comments:

Job method – please check the A values in Job plot (the values are not in accordance with the UV-Vis spectra)

The logK=8.52 for complex Cu+11   It is better to use another annotation (ß) for cumulative constant; the complex is 2:1.

Please, put the whole Table 1 on the same page.

Please give some comments on the purity of isolated 10+Cu complex.

Line 84: suggestion:  ….were synthesized according to the method developed in our research group

Author Response

The paper deals with synthesis of new calixarene derivatives as Schiff bases and their complexation properties towards Cu(II). Compounds were analyzed by usual analytical methods (NMR, IR, UV-Vis, CHN); stoichiometry and stability constants were determined by UV-Vis spectroscopy, selectivity was achieved regarding Ni(II), Co(II) and Zn(II) metal ions; results are well discussed and adequate conclusions are given.

I recommend the publication after minor revision.

Answer:

Dear Reviewer! Thank you very much for carefully reading and reviewing our paper!

Some comments:

Job method – please check the A values in Job plot (the values are not in accordance with the UV-Vis spectra)

Answer:

The images of the Job’s plots in Figures 3B, S21-S24 have been corrected.

The logK=8.52 for complex Cu+11   It is better to use another annotation (ß) for cumulative constant; the complex is 2:1.

The binding constant of complex Cu+11 has been denoted as K'b

Please, put the whole Table 1 on the same page.

Answer:

The caption and the main part of the Table 1 have been moved to one page.

Please give some comments on the purity of isolated 10+Cu complex.

Answer:

The purity of the complex was confirmed by elemental analysis.

Line 84: suggestion:  ….were synthesized according to the method developed in our research group

Answer:

The text has been corrected.

Reviewer 2 Report

The current piece of work is appropriate for Molecules - MPDI. However, please consider the following revision before publish:

  • In general, small errors in the use of English should be checked.
  • I recommend that you review the IUPAC (red book) recommendations regarding the nomenclature of compounds and how to refer to transition metal ions. 

Author Response

The current piece of work is appropriate for Molecules - MPDI. However, please consider the following revision before publish:

  • In general, small errors in the use of English should be checked.

Answer:

Dear Reviewer! Thank you very much for carefully reading and reviewing our paper! We have revised the manuscript throughout for errors, and to the best of our knowledge, we have corrected them.

  • I recommend that you review the IUPAC (red book) recommendations regarding the nomenclature of compounds and how to refer to transition metal ions. 

Answer:

Corrections of the names of compounds and their complexes with metal cations have been made in accordance with the recommendation of the reviewer.